# Robust data storage in DNA by de Bruijn graph-based de novo strand assembly

Lifu Song [1,2], Feng Geng[3], Zi-Yi Gong [1,2], Xin Chen[4], Jijun Tang[5,6], Chunye Gong [7], Libang Zhou[8], Rui Xia [7], Ming-Zhe Han [1,2], Jing-Yi Xu [1,2], Bing-Zhi Li [1,2] ✉ & Ying-Jin Yuan [1,2] ✉

DNA data storage is a rapidly developing technology with great potential due to its high density, long-term durability, and low maintenance cost. The major technical challenges include various errors, such as strand breaks, rearrangements, and indels that frequently arise during DNA synthesis, amplification, sequencing, and preservation. In this study, a de novo strand assembly algorithm (DBGPS) is developed using de Bruijn graph and greedy path search to meet these challenges. DBGPS shows substantial advantages in handling DNA breaks, rearrangements, and indels. The robustness of DBGPS is demonstrated by accelerated aging, multiple independent data retrievals, deep error-prone PCR, and large-scale simulations. Remarkably, 6.8 MB of data is accurately recovered from a severely corrupted sample that has been treated at 70 °C for 70 days. With DBGPS, we are able to achieve a logical density of 1.30 bits/cycle and a physical density of 295 PB/g.

DNA is the natural solution for the preservation of the genetic information of all life forms on earth. Recently, million years old genomic DNA of mammoths was successfully decoded, revealing its great potential as a long-term data carrier under frozen conditions[1]. Owing to its high density and low maintenance cost, revealed by recent studies, DNA has been considered as an ideal storage medium to meet the emerging challenge of data explosion[2–20]. Frequently occurring errors in DNA synthesis, amplification, sequencing, and preservation, however, challenge the data reliability in DNA. Efforts to solve these issues led to the implementation of a codec system that requires two layers of error correction (EC) codes. The outer layer codes handle the strand dropouts and the inner layer codes deal with the intramolecular errors, ensuring accurate data readouts[21–25]. The design of outer codes is straightforward since strand dropouts can be well solved by sophisticated erasure codes, e.g. Fountain or Reed-Solomon (RS) codes[21–23]. In contrast, the design of an inner codec system is challenging due to the complex errors and the unique feature of 'data reputations', i.e., the

noisy strand copies[10,21,26]. The traditional decoding process of inner codes generally consists of two steps: clustering and strand reconstruction[23,26,27]. Strand reconstruction from its error-rich copies is a trace reconstruction problem introduced two decades ago[28]. Most studies on trace reconstruction are motivated by the multiple-alignment problem in computational biology[23,28–34]. The error types of substitutions and indels have been widely investigated by previous studies on trace reconstruction, which reveal the difficulties of handling indels[23,28–34].

In practice, DNA breaks and rearrangements occur frequently during the preservation of DNA molecules and polymerase chain reaction (PCR) based data copying, threatening the robustness of DNA data storage. Under certain conditions and in long-term storage, DNA is subjected to hydrolysis and degradation which lead to DNA breaks. This highlights the importance of data stability studies on the influences of harsh conditions, such as high temperature or UV exposure, and methods of DNA protection[21,35,36]. Unspecific amplification, a

[1]Frontiers Science Center for Synthetic Biology and Key Laboratory of Systems Bioengineering (Ministry of Education), Tianjin University, Tianjin 300072, China. [2]School of Chemical Engineering and Technology, Tianjin University, Tianjin 300072, China. [3]College of Pharmacy, Binzhou Medical University, Yantai 264003 Shandong Province, China. [4]Centor for Applied Mathematics, Tianjin University, Tianjin 300072, China. [5]School of Computer Science and Technology, College of Intelligence and Computing, Tianjin University, Tianjin 300350, China. [6]Shenzhen Institute of Advanced Technology, Chinese Academy of Sciences, Shenzhen, China. [7]National SuperComputer Center in Tianjin, Tianjin 300457, China. [8]College of Food Science and Technology, Nanjing Agricultural University, Nanjing 210095 Jiangsu Province, China. ✉e-mail: bzli@tju.edu.cn; yjyuan@tju.edu.cn

typical problem with PCR, is the main source of DNA rearrangements, which refer to the breakage and rejoining of DNA strands in DNA data storage. People currently carefully design primers and optimize PCR conditions to avoid unspecific amplification, and employ gel purification to physically exclude rearranged strands from the data pool[23]. An inner codec system that can tolerate fragmented and rearranged DNA strands is nevertheless critical for enhancing the robustness of DNA data storage. The clustering[27,37] and multiple-alignment[38–40] (CL-MA) algorithms, however, are both incapable of dealing with DNA breaks and rearrangements. A new mechanism that can efficiently handle DNA breaks and rearrangements is highly desirable.

In this study, we propose a de novo assembly-based strategy to handle the complex errors, especially the DNA breaks and rearrangements, in DNA data storage channel. Different from the CL-MA-based methods, we first decompose all the strand sequences into $k$-mers with the de Bruijn graph (DBG) theory[41–44]. Then, the low-occurrence $k$-mers are excluded to omit huge errors. After that, the strands are assembled straightforwardly through greedy path search and path selection with the aid of the embedded redundancy codes. Compared with the CL-MA-based methods, this DBG-based greedy path search algorithm (DBGPS) shows substantial advantages in the handling of errors, especially DNA breaks, rearrangements, and indels. The effectiveness of DBGPS with large datasets is verified through large-scale simulations up to 1 GB (Gigabytes, $10^9$ Bytes). The robustness of DBGPS is demonstrated by three harsh experiments of

accelerated aging, multiple data retrievals, and deep error-prone PCR with 6.8 MB (Megabytes, $10^6$ Bytes) input data. Remarkably, we are able to precisely retrieve the entire 6.8 MB data from a DNA solution that has been incubated at 70 °C for 70 days without any particular protection using DBGPS. Besides the high data robustness, we were able to achieve a high logical density of 1.3 bits per synthesis cycle and a high physical density of 295 PB (Petabytes, $10^{15}$ Bytes) per gram of DNA with DBGPS.

## Results

### Design of DBGPS: a de novo assembly-based strand reconstruction algorithm

The DNA data storage channel is a complex channel with several types of errors. Most previous studies have focused on substitution and indel errors[28,32,36,45–48]. In practice, however, DNA breaks and rearrangements occur frequently during the preservation of DNA molecules and PCR-based data copying, threatening the robustness of DNA data storage. These two types of errors should be considered in the DNA data storage channel (Supplementary Fig 1a). However, DNA breaks and rearrangements can cause severe failures to the traditional EC algorithms. To handle these issues, as illustrated in Fig. 1a, we focus on a distinct route of de novo assembly which potentially takes advantage of the "multi-copy" feature of DNA for accurate strand reconstruction. The successful applications of de Bruijn graph (DBG) in genome assembly[41–43] prompted us to investigate its potential along this route.

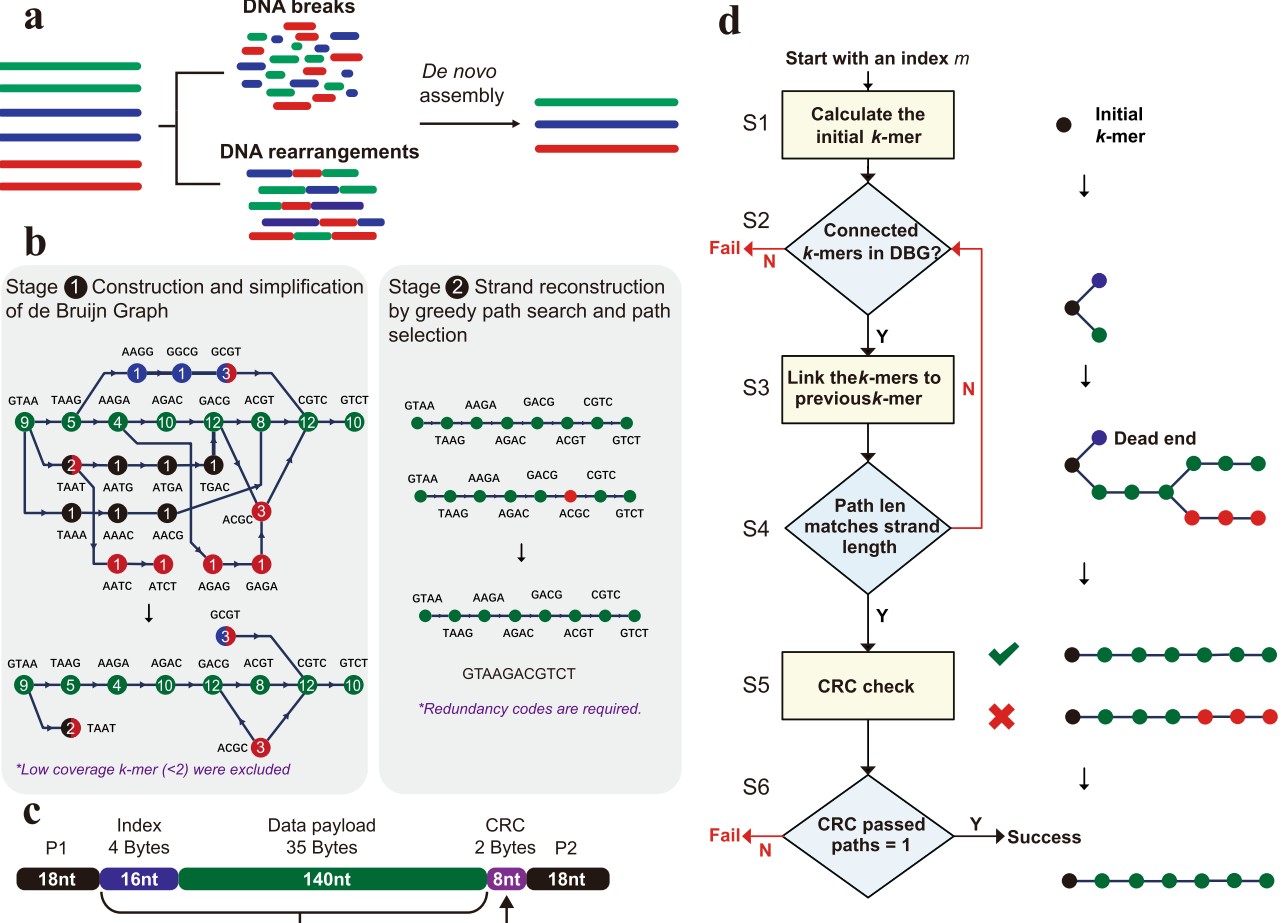

**Fig. 1 | De novo assembly-based strand reconstruction for DNA data storage. (a)** The issues of DNA breaks and rearrangements in DNA data storage and the proposed de novo assembly-based strategy for dealing with them. **(b)** The two-stage de novo assembly process of the proposed de Bruijn graph-based greedy path search algorithm (DBGPS). The representative de Bruijn graph in stage 1 was constructed from the nice error-rich sequence copies shown in Supplementary Fig 1a with a $k$-mer size of four. The circles stand for the $k$-mer nodes. The numbers inside the circles are the occurrences, *i.e.*, coverages, of corresponding $k$-mers. The correct sequence is represented by the path of green nodes. **(c)** The designed strand structure for the DBGPS algorithm. **(d)** The workflow of the greedy path search and selection process.

Since a DNA strand can be assembled only if the strand path is intact in DBG, we ran simulations to estimate the integrity probability of a strand path with variant strand copies containing various rates and types of errors. Here, this probability was defined as the theoretical maximal strand reconstruction rate ($S_m$) of DBG-based strand reconstruction. Strand copies in a range of 3 to 25 were considered, and the detailed simulation results are provided in Supplementary Data 1. As shown by the representative results obtained with five and ten-strand copies, the $S_m$ values, as expected, are inversely related to the error rates but are similar regardless of the error types at the same error rates (Supplementary Fig 1c). Importantly, with merely ten sequence copies, the sequence path in DBG shows high robustness even under a high strand error rate of 5%, proving the feasibility of DBG-based strand reconstruction. As illustrated in Fig. 1b, we then proposed the DBGPS algorithm for error-free reconstruction of strands, which comprises two stages as follows:

Stage 1, construction and simplification of DBG. The construction of DBG here refers to $k$-mer counting[49,50]. The connections between the $k$-mers do not need to be constructed. The $k$-mer counting data in a hash table is well-suited for the greedy path search step in Stage 2. Massive errors could cause enormous noise $k$-mers in DBG. The exclusion of noise $k$-mers is crucial for reducing the computing complexity in Stage 2. The occurrences of specific errors are low-probability events individually. Therefore, the occurrences, i.e., coverages, of noise $k$-mers are generally lower than those of the correct ones. Monte Carlo simulations indicate that coverage is a suitable indicator for the exclusion of noise $k$-mers (Supplementary Fig. 2). The $k$-mer size is an important parameter for strand reconstruction. With a specific $k$-mer size, there is a theoretical upper boundary to the volume of data that can be decoded. For example, only several bits of data can be decoded with a $k$-mer size of three. In DBG theory, for each $k$-mer, the front $k-1$ bases were used for positioning, and the terminal base was used for path extension, i.e., encoding of fresh data. The greedy path search step in Stage 2 requires each front $k-1$ base combination to ideally be present once in DBG to avoid path loops and forks. Based on this principle, the decoding capacity of DBGPS with a $k$-mer size of 27 was estimated to be around 1 TB (Terabytes, $10^{12}$ Bytes). The detailed choices of $k$-mer sizes associated with data volumes ranging from 1 KB (Kilobytes, $10^3$ Bytes) to 1 EB (Exabytes, $10^{18}$ Bytes) are listed in Supplementary Table 1. More details of the estimation process are provided in the Methods of the Supplementary Information.

Stage 2, strand reconstruction by greedy path search and path selection. The strand reconstruction process is achieved by greedy path search for strand candidates and path selection using the embedded EC codes for the correct strand. To facilitate this process, as shown in Fig. 1c, we designed a strand structure that contains 16 nt index, 140 nt data payload, and 8 nt Cyclic Redundancy Check (CRC) code, flanked by landing sites for sequencing primers. As illustrated in Fig. 1d, six key steps are required to reconstruct the strand sequence of a specific index $m$. Step 1 (S1), encode the index $m$ into a DNA string and calculate the initial $k$-mer. This initial $k$-mer is also the first terminal $k$-mer. Step 2 (S2), check if there are connected $k$-mers for each terminal $k$-mer. The terminal $k$-mers without connected $k$-mers are marked as dead-ends. Strand reconstruction fails if all the terminal $k$-mers are dead-ends. Step 3 (S3), connect all connected $k$-mers to the corresponding terminal $k$-mers. Step 4 (S4), check if the path length matches the strand length. If not, go to S2. If so, go to S5. Step 5 (S5), perform parity check for each path candidate using the embedded CRC codes. Step 6 (S6), check if only precisely one path passes the CRC check and select the only path as the correct path, i.e., the strand sequence of index $m$. Strand reconstruction fails if multiple or no paths pass the CRC check. DBGPS will continue to assemble the next strand of index ($m + 1$) until all possible indexes are processed.

## The error handling capability and large data scale performance of DBGPS

As shown in Fig. 2a–e, we ran simulations to estimate the error handling capability of DBGPS in comparison with the multiple-alignment (MA) algorithm using twenty strand copies. Remarkably, as shown in Fig. 2a and b, DBGPS shows high performance in handling DNA breaks and rearrangements. In contrast, the strand decoding rate of MA declined to zero rapidly with the introduction of only a small number of DNA breaks and rearrangements. DBGPS also shows a clear advantage in handling indels, especially when the error rate is high (Fig. 2c). In the case of substitutions (Fig. 2d), DBGPS shows a decrease in strand decoding rates when high rates of substitutions are introduced. It has been reported that even with the low-quality synthesis method of light-directed synthesis, the overall error rate was estimated to be around 6%[51]. Thus, the slight deficiency of DBGPS in the handling of high rates of substitutions would not hamper its practical application in DNA data storage. Since the errors are mixtures of different types of errors in practice, we then ran simulations with mixed errors, which include substitutions, indels, DNA breaks, and rearrangements in a ratio of 1:1:2:1:1 respectively. As expected, compared with MA, DBGPS shows substantial advantages in handling of mixed errors, as shown in Fig. 2e. Next, to investigate how the strand copy number affects the performance of DBGPS and MA, we ran simulations with various strand copies with a fixed error rate of 3% (1.5% substitutions, 0.75% insertions, and 0.75% deletions). As shown in Fig. 2f, DBGPS achieves a higher $S_r$ value than MA in general, except for the cases with extremely low copy numbers below six. The strand reconstruction rates ($S_r$) of DBGPS and MA both increase significantly by introducing more strand copies. With more strand copies introduced, the $S_r$ value of DBGPS increases more quickly and surpasses the value of MA at the point of seven. This suggests that DBGPS utilizes the "multi-copy" feature more effectively than MA to enhance the data robustness.

To evaluate the performance of DBGPS with large data volumes, a series of simulations were performed with data sizes ranging from 1 MB to 1 GB. For each data size, three independent simulations were performed using random seeds of 1, 2 and 3 for the generation of the DNA droplets/strands. Preliminary simulations at the GB level showed a significant increase in reconstruction time per strand, which was caused by entanglements of strand paths in DBG, i.e., the repeated presentation of $k$-mers in different strands. To solve this problem, a strand filtering process, as illustrated in Supplementary Fig 3, was designed and applied to filter out the entangled strands. The DNA strand sequences after filtering were utilized for the generation of error-rich copies. Error-rich strand sequences were simulated with a copy number of 25 and an error rate of 3% (1.5% substitutions, 0.75% insertions, 0.75% deletions). Counting of $k$-mer has previously been demonstrated to be a linear problem[49,50,52,53], and this finding was confirmed in this study (Supplementary Fig. 4). For strand reconstruction by DBGPS, the reconstruction time per strand increased slowly when the data size scaled up from 1 MB to 1 GB (Fig. 2g). A time complexity of $O(nlogn)$, where $n$ stands for data size, was revealed by a fitting experiment (Supplementary Fig 5). Importantly, no significant difference in decoding accuracy was observed with the data volumes ranging from 1 MB to 1 GB (Supplementary Fig. 6a). The DBG-based sequence reconstruction has been widely studied to solve the genome assembly problem, and such studies reveal the challenge of accurate sequence reconstruction by DBG[41,42,54]. It should be noted that the sequence reconstruction problem in DNA data storage is significantly different from the genome assembly problem. Rather than long sequences, the DNA sequences in DNA data storage are short fragments with a fixed length of 100-300 bp. As illustrated in Fig. 2h, the DBG constructed with short fragments shows substantial differences in structure from those that are constructed with long sequences. The nodes, i.e., the $k$-mers, of DBG derived from long sequences are tightly connected. In contrast, the nodes in the DBG of short DNA

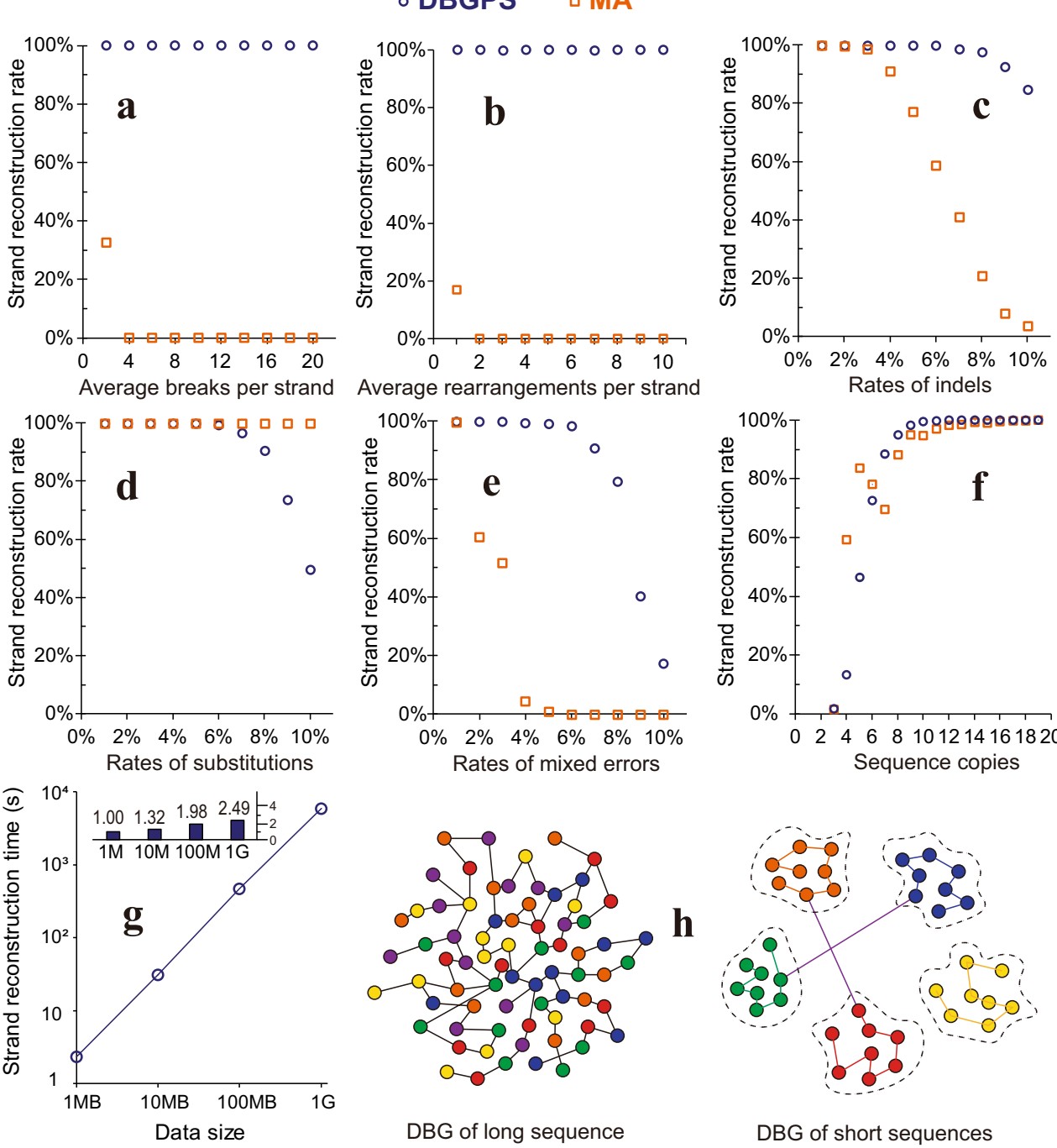

**Fig. 2 | Error-handling capabilities of DBGPS in comparison with MA and large-scale performance simulations.** With twenty sequence copies, the performance of DBGPS and MA in handling various rates of (**a**) DNA breaks, (**b**) DNA rearrangements, (**c**) indels, (**d**) substitutions, and (**e**) mixed errors. The mixed errors comprise DNA breaks, DNA rearrangements, substitutions, insertions, and deletions in a ratio of 1:1:2:1:1. **f** Strand reconstruction rates with various strand copies containing 3% error mixtures of substitutions (1.5%), insertions (0.75%), and deletions (0.75%). **g** Strand reconstruction time by DBGPS with data scales ranging from 1 MB to 1 GB. The small bar chart at the top shows the fold changes in reconstruction time per strand compared to that of the 1 MB scale. (**h**) Illustration of the differences between the DBG constructed with numerous short sequences and that constructed with long sequence(s). The nodes with different colors stand for various *k*-mers. Each node stands for a unique *k*-mer. Data are presented as mean values of three independent simulations in figures **a**–**g**. The standard deviation (SD) values, which are too small to be clearly visualized, are listed in the source data. Source data are provided as a Source Data file.

sequences are spontaneously separated by *k*-mer editing distance. This was proved by the analysis of the cross-linked strands with data volumes ranging from 1 MB to 1 GB, as shown in Supplementary Fig 6b. Importantly, these cross-linked strands can be eliminated by the filtering process illustrated in Supplementary Fig 3, enabling the high efficiency of DBGPS with large datasets. Furthermore, the short fragments in DNA data storage are indexed and are embedded with EC codes, which guarantee the accurate assembly of the original information.

**Experimental verification of the robustness of DBGPS**
As shown in Fig. 3a, ten digital pictures of Dunhuang murals in a 6.8 MB zipped file (6,818,623 bytes, Supplementary Data 1) were encoded into 210,000 DNA strands of 200 bp (Supplementary Data 2) with a

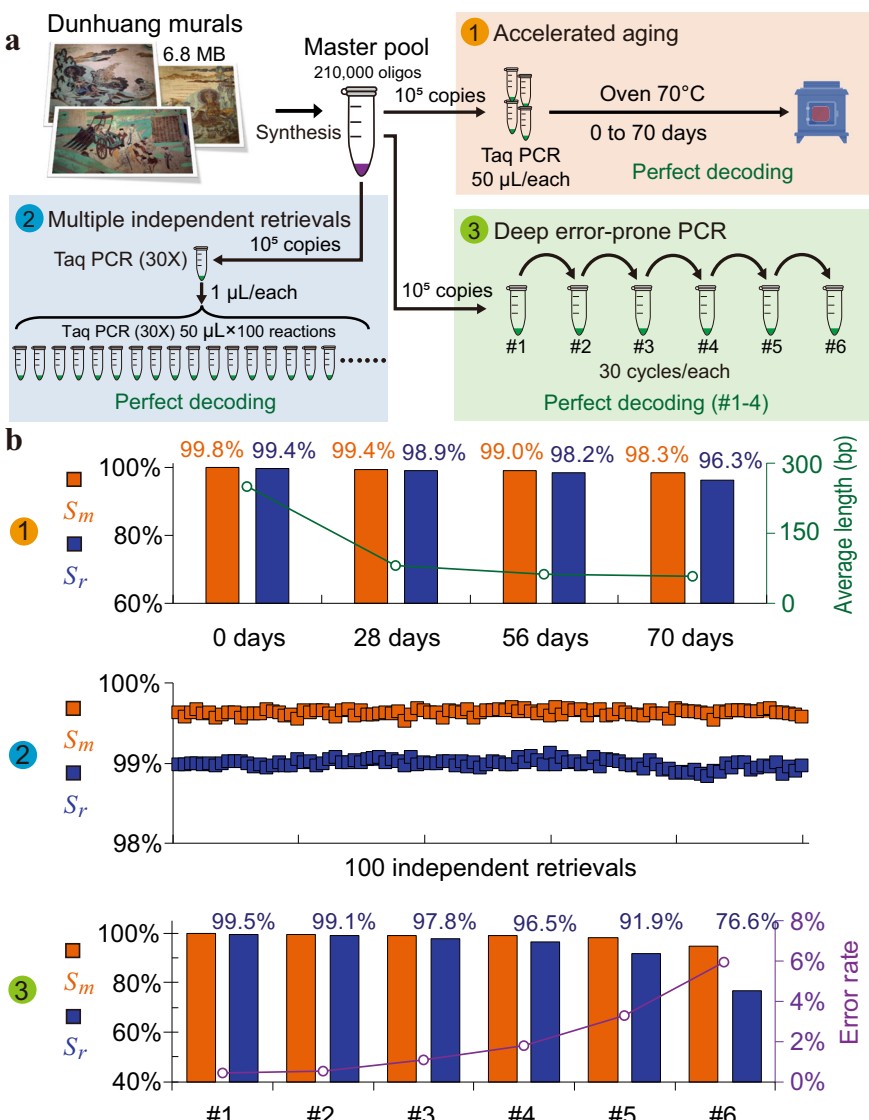

**Fig. 3 | Experimental verification of the robustness of DBGPS. a** Illustration of the three harsh experiments (1, 2, 3) performed to verify the robustness of DBGPS. A 6.8 MB zipped file of Dunhuang mural pictures was recorded by oligo synthesis, generating an ssDNA "Master Pool" with 210,000 unique types of ssDNA strands. Experiment 1, accelerated aging to verify the robustness with DNA degradations (breaks). Experiment 2, multiple data retrievals with intended unspecific amplifications to verify the robustness with strand rearrangements. Experiment 3, deep error-prone PCR to introduce errors. **b** Data retrieval details of the three experiments. $S_m$ stands for maximal strand recovery rate. $S_r$ stands for strand recovery rate. The green curve shows the average fragment lengths of the accelerated aging samples. The purple curve shows the error rates of the deep error-prone PCR samples. The strand reconstruction details of the three experiments are provided in Supplementary Table 3, 4 and 5 respectively. The Dunhuang mural pictures were obtained from Dunhuang Academy (http://www.dha.ac.cn/) with permission for this study. All rights reserved for other uses.

structure as shown in Fig. 1c. The strand redundancy was set to 7.8%, which supports reliable data recovery when the decoder receives more than 95% of strands. The designed DNA strands were synthesized by Twist Bioscience and the obtained oligos were dissolved in ddH$_2$O to generate a "master pool". Three harsh experiments were performed with this "master pool" to obtain low-quality samples with various rates and types of errors. The low-quality samples obtained were then sequenced on the Illumina sequencing platform. The raw reads generated were processed by DBGPS for strand reconstruction followed by decoding using the outer fountain codes.

In silico simulation has shown that DBGPS handles high rates of DNA breaks well. Accelerated aging experiments were performed to further confirm its practical performance in tolerating DNA degradation. The purified PCR products in elution buffer were incubated at 70 °C for a prolonged period of 0, 28, 56, and 70 days. As shown by the Agilent 2100 Bioanalyzer analysis

(Supplementary Fig 7), the integrity of the DNA strands (200 bp) has been severely damaged after incubation at 70 °C for merely 28 days, as confirmed by the sequencing results (Supplementary Fig 8). Such a degree of degradation made it impossible to decode the original information using CL-MA (Supplementary Table 2). By contrast, DBGPS achieves high $S_r$ values from all accelerated aging samples, ensuring accurate data retrievals (Fig. 3b and Supplementary Table 3). Notably, a high $S_r$ value of 96.3% is even obtained with the sample that has been treated at 70 °C for 70 days. Data retrievals from accelerated aged DNAs have been reported[21,35,55]. In these studies, the authors tested the effects of different methods for the preservation of DNA molecules, among which embedment in silicon beads was shown to be the best, allowing recovery of information after treatment at 70 °C for 1 week. According to the authors, this was thermally equivalent to storing information on DNA at 9.4 °C for 2,000 years[21]. Based on

**Table 1 | Key achievements of this work in comparison with prior DNA storage studies**

|  | Church et al.[3] | Goldman et al.[24] | Grass et al.[21] | Erlich et al.[22] | Organick et al.[23] | Leon et al.[57] | Antkowiak et al.[26] | This work |
|---|---|---|---|---|---|---|---|---|
| Data size (MB) | 0.53 | 0.74 | 0.08 | 2.14 | 200.2 | 6.42 | 0.1 | 6.8 |
| Total oligos | 54,898 | 153,335 | 4991 | 72,000 | 13,400,000 | 217,000 | 16,383 | 210,000 |
| Logical density (Bits/cycle) | 0.6 | 0.19 | 0.86 | 1.19 | 0.81 | 1.52 | 0.94 | 1.30 |
| Physical density (PB/g) | NA | NA | NA | 215 | NA | 5.9 | NA | 295 |
| Long-term stability (9.4 °C) | – | – | ~2000 years | – | – | – | – | ~20,000 years |

The long-term stability was estimated based on the results shown in Fig. 3 and the study by Grass et al.[21]. Primers were considered in the calculation of logical density (bits per synthesis cycle). Strand reconstruction details of the high-density storage study in this work is provided in Supplementary Table 6.

this estimation, the DBGPS algorithm can retrieve data accurately from DNA solutions preserved at 9.4 °C without any protection for 20,000 years.

Copying and retrieval of data stored in DNAs require PCR-based amplification. As a typical biological process, unspecific amplification occurs occasionally, leading to DNA rearrangements. To demonstrate the effectiveness of DBGPS in the handling of DNA rearrangements, 100 independent data retrievals with PCR-amplified products were performed (Fig. 3a), in which unspecific amplifications were intendedly introduced. The CL-MA-based method shows low $S_r$ values of less than 12% with three representative samples (Supplementary Table 2). In contrast, high $S_r$ values ranging from 98.8% to 99.1% were obtained with DBGPS, which ensure perfect data recovery in all retrievals (Fig. 3b, Supplementary Table 4).

A series of six error-prone PCR (ePCR) amplifications were performed to introduce large numbers of base errors at various rates. All ePCR reactions were performed for 30 thermal cycles. Although the rough amplification conditions introduced massive errors as expected, strikingly high $S_r$ values in a range of 96.6% to 99.5% were obtained for ePCR#1-4, ensuring accurate data recovery (Fig. 3b, Supplementary Table 5). More errors are expected to introduce more branch paths in DBG, raising the necessary cutoff for noise $k$-mer elimination. Consequently, more correct $k$-mers were consequently discarded by mistake, resulting in lower $S_r$ values. As expected, declines in $S_r$ were observed as more rounds of ePCR were performed. Although the data retrievals with ePCR#5 and ePCR#6 failed, the total 120 cycles of amplification performed in ePCR#1-4 can already guarantee sufficient reliable data copies.

## Discussion

DNA data storage technology maintains the order of information through defined sequences of nucleic acid chains. Different from the traditional plane media, which require a surface to maintain the information order, the nucleic acid chains can be detached from the writing surface and distributed in a three-dimensional space without affecting the data integrity. This gives DNA data storage a huge increase in data density. However, the chain issues of breakages and incorrect linkages, i.e., DNA breaks and rearrangements, need to be addressed. As illustrated in Supplementary Fig 9, in a radical view, substitutions and indels can also be regarded as special cases of "DNA rearrangements", which implies the importance of these fundamental issues. However, DNA breaks and rearrangements are difficult to tackle using traditional EC codes. In this study, we developed DBGPS, a de novo assembly-based strand reconstruction algorithm based on DBG theory, which takes advantage of the unique "multicopy" feature for the error-free reconstruction of strand sequences. We verified the effectiveness of DBGPS in the handling of DNA breaks and rearrangements, as well as substitutions and indels, through both dry and wet experiments. With this algorithm, we were able to reconstruct DNA strands accurately from error-rich strand copies that were mixed together without a clustering step.

The most significant contribution of this work is the strategy that addresses the data reliability issues caused by DNA degradation. The importance of the DNA degradation problem in DNA data storage has been well discussed in a recent review[36]. Different from the previous studies, which focused on protecting the DNA molecules from corruption[21,35,36,55], here we focused on the implementation of a computational process to reconstruct the strand sequences even if the strands are broken into small fragments of dozens of bases. As proved by the accelerated aging experiment, we effectively addressed the data robustness issue caused by DNA degradation with DBGPS. It is worth noting that the encoded information from a DNA solution that has been incubated at 70 °C for 70 days without any particular protection can be accurately recovered with our algorithm. The degree of damage to DNA under this heating condition was estimated to be equivalent to that under 9.4 °C for more than 20,000 years[21]. This period is already longer than the oldest written record of human civilization, the Cuneiform, which dates back about 5,500 years ago[56]. Such unprecedented robustness highlights the importance of our method and suggests the great potential of our DBGPS algorithm in DNA data storage. DBGPS is presumably compatible with a variety of DNA preservation methods, e.g., silicon beads[21], nanoparticles[55] and alkaline salts[35], to further enhance data stability. Altogether, it is reasonable to believe that DNA data storage technology would allow us to preserve human civilizations for a very long time, even after the unavoidable destruction of mankind in the distant future. Another significant contribution of this study is the efficient handling of unspecific amplification. Retrieval of DNA data requires PCR-based amplification of strands. As a typical biological process, PCR has stochastic feature naturally. This stochastic feature frequently leads to unspecific amplification, threatening the robustness of DNA data storage. With DBGPS, we were able to achieve reliable data retrieval even when unspecific amplification occurs, as proved by the 100 independent data retrievals with severe unspecific amplification intentionally introduced.

Based on traditional information theory, redundancy data in the form of "repetition" is unfavorable since the production of additional data copies on traditional media is cost-ineffective and time-consuming. By contrast, data repetition in the form of DNA strand copies can be obtained inexpensively and speedily. Thus, the authors believe that the traditional definition of "coding efficiency" needs to be adapted to the DNA data storage channel. With current technologies, producing a single molecular copy of DNA data is not even achievable. It has been reported that serial dilutions to reduce the copy numbers of information-bearing DNA strands till the average copy number is below ten result in massive strand dropouts, making reliable data retrieval impractical[46]. Even if a single molecular copy of DNA data is obtainable, it may be damaged due to DNA degradation under certain conditions or during long-term storage. Taken together, the design for DNA storage should be based on reality and necessity, allowing for the presence of more than one copy of DNA strands and not too many copies to maintain high information density. In other words,

"multicopy" is crucial for the robustness of DNA data storage and should ideally be maintained in a reasonable range. Data robustness, physical density, and economy together determine the technical advancements of a storage medium. By taking advantage of the "multicopy" feature, this work significantly improved the robustness of DNA data storage while maintaining high physical and logical density. The latter is an important indicator highly related to writing costs. As detailed in Table 1 and Supplementary Table 6, we achieved a physical density of 295 PB/g and a logical density of 1.30 bits/cycle at a data scale of 6.8 MB. Recently, Anavy et al. have reported a logical density of 1.53 bits/cycle using composite DNA letters at a scale of 6.4 MB. However, a low physical density of 5.9 PB/g is reported due to the high coverage required by composite DNA letters[57]. Compared to the study of Anavy et al., we achieve much higher physical density and data robustness at the cost of a slight decrease in logical density. It's worth mentioning that we obtained a strand dropout rate of 1.79% at a scale of 6.8 MB, which is significantly lower than the reported dropout rate of 3.60% at a scale of 2.14 MB by Erlich et al., revealing the technical advances of DBGPS.

Although the mechanism of low coverage $k$-mer exclusion works well with most of the data retrievals in this study, we observed a significant decrease in strand reconstruction rate when more and more errors are introduced (Fig. 3b, Supplementary Table 5). Future incorporation of redundancy codes that can help to identify and exclude the noise $k$-mers is expected to further improve the efficiency of DBGPS, which is particularly important in the extreme case of high error rates[26,51]. The recent study by Antkowiak et al. revealed the potential of low-quality synthesis methods, e.g., photolithographic, and electrochemical synthesis, in reducing the writing cost of DNA data storage. However, due to the high error rate of DNA synthesis, a low strand recovery rate of 83% was reported using the CL-MA method[26]. Interestingly, DBGPS combined with a simple mechanism of inserting one error-checking base every few bases (block checking codes)[58] could be a potential solution to this problem. With block lengths ranging from 3 to 10, simulation tests of this strategy on the sequencing data from the study of Antkowiak et al.[26] revealed high strand reconstruction rates ranging from 98.3% to 99.6% (Supplementary Table 7), suggesting a potential strategy in low-cost, inaccurate DNA synthesis technologies for data storage.

### Reporting summary
Further information on research design is available in the Nature Research Reporting Summary linked to this article.

## Data availability
Source data are provided with this paper. The sequencing data generated in this study have been deposited in the figshare database under the following DOI links:

Accelerated aging samples of 70 °C for 0 and 28 days. https://doi.org/10.6084/m9.figshare.17193170.v2 Accelerated aging samples of 70 °C for 56 and 70 days. https://doi.org/10.6084/m9.figshare.17192639.v1 Three samples of the 100 independent retrievals. https://doi.org/10.6084/m9.figshare.18515078.v1 Error-prone PCR 1st and 2st rounds. https://doi.org/10.6084/m9.figshare.16727122.v2 Error-prone PCR 3st and 4st rounds. https://doi.org/10.6084/m9.figshare.17193128.v1 Error-prone PCR 5st and 6st rounds. https://doi.org/10.6084/m9.figshare.18515045.v1 High density storage – 295PB/g. https://doi.org/10.6084/m9.figshare.17183081.v1 Source data are provided with this paper.

## Code availability
All the original algorithms proposed in this study were implemented with Python and the source codes are released on Zenodo at https://doi.org/10.5281/zenodo.6833784[59]. A compiled C implementation of DBGPS is available at https://doi.org/10.5281/zenodo.6833747[60]. The source codes of the C version can be obtained for academic usage upon request to the authors.

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

## Acknowledgements

This work was supported by Natural Science Foundation of China under the grants 21621004 to YJY, Seed Foundation of Tianjin University to LFS. The authors would like to thank Dr. Yan Zhang for his great efforts and constructive comments which have helped to improve this manuscript significantly. The authors also would like to thank Dr. Feng Gao, Dr. Hao Qi, Dr. Weigang Chen, Dr. Yu Lin, and Dr. Sheng Ye for their insightful discussions. The authors also would like to thank Dunhuang Academy for providing the digital images of Dunhuang murals.

## Author contributions

L.F.S. and Y.J.Y. conceived the study. L.F.S. proposed and designed the algorithm, participated in the experiment design and data analysis. F.G. performed the performance testing and participated in the algorithm and data analysis. Z.Y.G. performed all the experiments. C.Y.G. and R.X. participated in the implementation of DBGPS-C. J.J.T. and X.C. participated in the decoding complexity analysis. L.B.Z., M.Z.H. and J.Y.X. helped with the experiments. B.Z.L. and Y.J.Y. edited the manuscript and supervised the whole work.

## Competing interests

LFS, YJY and FG have granted a patent covering the encoding structure and the decoding process (ZL 2019 1 0465200.7). The remaining authors declare no competing interests.
