## [Peer Review File · Nature Communications]

Reviewers' Comments:

Reviewer #1:

Remarks to the Author:

Using synthetic DNA molecules for storing digital data is an emerging storage technology that has received a lot of recent attention in the scientific literature. Almost all existing papers model errors that occur in short DNA strands during DNA synthesis, storage, and sequencing, as insertions, deletions, and substitutions. The current paper proposes a more general model for errors that is also well justified by practice. In particular, the authors also consider DNA strand breaks and strand re-arrangements. The main contribution of the paper is a new decoding method based on de Bruijn graphs, that (unlike methods from the literature) is well suited to handle strand breaks and rearrangements, apart from the other types of errors. De Bruijn graphs have been used in the context of genome assembly, but to the best of the knowledge of the current reviewer this is the first paper that suggests using these methods in the context of DNA data storage. The authors make the source code they use publicly available. I find the paper interesting.

Some minor comments:

1. Notation S_m use don line 104 has not been defined. What is m ?
2. Line 126: "Terabyte" is usually written without a space in the middle. Same goes foe "Exabyte" and "Kilobyte".

Reviewers' comments:

Reviewer #1:

Reviewer #1 (Remarks to the Author):

Using synthetic DNA molecules for storing digital data is an emerging storage technology that has received a lot of recent attention in the scientific literature. Almost all existing papers model errors that occur in short DNA strands during DNA synthesis, storage, and sequencing, as insertions, deletions, and substitutions. The current paper proposes a more general model for errors that is also well justified by practice. In particular, the authors also consider DNA strand breaks and strand re-arrangements. The main contribution of the paper is a new decoding method based on de Bruijn graphs, that (unlike methods from the literature) is well suited to handle strand breaks and rearrangements, apart from the other types of errors. De Bruijn graphs have been used in the context of genome assembly, but to the best of the knowledge of the current reviewer this is the first paper that suggests using these methods in the context of DNA data storage. The authors make the source code they use publicly available. I find the paper interesting.

The authors appreciate the positive evaluation of this work.

Some minor comments:

Notation S_m use don line 104 has not been defined. What is m ?

Reply:

Notation S_m stands for “maximal reconstruction rate of strands”. Specifically, symbol “ S ” stands for “strands” and symbol “ m ” stands for “maximal reconstruction rate”.

2. Line 126: “Terabyte” is usually written without a space in the middle. Same goes for “Exabyte” and “Kilobyte”.

Reply:

Thanks, and we have revised accordingly.